# Self-pretrained V-net based on PCRL for Abdominal Organ Segmentation

Jiapeng Zhang

University of Shanghai for Science and Technology, Shanghai, China

**Abstract.** Abdomen organ segmentation has many important clinical applications. However, the manual annotating process is time-consuming and labor-intensive. In the "Fast and Low-resource semi-supervised Abdominal oRgan sEgmentation in CT" challenge, the organizer provide massive unlabeled CT images. To effectively utilize unlabeled cases, we propose a self-pretrained V-net. Inspired by the preservational contrastive representation learning (PCRL), the proposed method consists of two steps: 1) using a large amount of unlabeled data to obtain a pre-trained model, 2) using a small amount of labeled data to perform fully supervised fine-tuning on the basis of the former. The feature extraction part used in both stages uses the same backbone network. The difference is that the pre-training stage introduces the additional image reconstruction branch and the corresponding momentum branch to construct image reconstruction and contrastive learning, and the fully-supervised model downstream uses a fully convolutional network for segmentation prediction. In the pre-training stage, by incorporating diverse image reconstruction tasks into the contrastive learning, the representation ability of the backbone network for specific image data during the upstream feature extraction process is enhanced. Besides, the half-precision (Float16) is used in the prediction stage, which reduces the GPU load by about 36% without losing the prediction accuracy and the maximum used GPU memory is 1719 MB. Quantitative evaluation on the FLARE2022 validation cases, this method achieves the average dice similarity coefficient (DSC) of 0.4811 and average normalized surface distance (NSD) of 0.4513.

**Keywords:** Self-supervised learning · Self-transfer learning · Organ Segmentation.

## 1 Introduction

Abdominal organ segmentation plays an important role in clinical practice, the state-of-the-art methods have achieved inter-observer performance in several benchmark datasets. However, most of the existing abdominal datasets only contain single-center, single-phase, single-vendor, or single-disease cases, and it is unclear whether the excellent performance can be generalized on more diverse datasets. Some SOTA methods have good general applicability. However, when the training data is limited and the task is complex, it is difficult for the model to

be fully trained. Moreover, many SOTA methods use model ensembles to boost performance, but these solutions usually have a large model size and cost extensive computational resources, which are impractical to be deployed in clinical practice.

Compared with labeled data, unlabeled data is usually easier to obtain because the manual labeling process is omitted. To make full use of the massive unlabeled cases, self-supervised learning has been widely adopted[2]. Based on the massive unlabeled data provided by the Fast and Low-resource semi-supervised Abdominal oRgan sEgmentation in CT challenge, we attempted to design our method based on V-Net[7], and PCRL[12].

Specifically, the backbone uses the encoder-decoder style architecture with skip connection [8]. The vast majority of successful algorithms for image segmentation in the medical domain such as V-net [7] and Dense U-net [10] are based on this U-shape structure. For unlabeled data, we use the method of retaining contrastive representation learning to obtain a pre-training weight through self-supervised learning. Then, perform full supervision finetuning through limited annotated data. Note that this pre-trained model was trained from the unlabeled cases provided by the challenge, and no additional pre-trained models were used in the process. Compared with methods that only use contrastive learning, PCRL can generate stronger representations of image information in the upstream feature extraction network by reconstructing different contexts. Besides, to take into account the use of GPU memory and the preservation of information between multiple organs and backgrounds, we adopt a horizontal plane scaling and vertical sliding window strategy to train the model. Meanwhile, due to the limitation of GPU resources, we use a smaller input size to reduce resource consumption.

The main contributions of this work are summarized as follows:

1) We propose a PCRL-based self-pretrained multi-organ segmentation framework to make full use of the massive unlabeled cases.

2) To reduce resoure consumption and speed up the inference process, we compress the input size and utilize a smaller width for the network.

## 2   Method

As mentioned in Fig 1, this whole segmentation framework is composed of a selfsupervised pretrain stage and a full-supervised finetuning stage. The detail description of the method is as follows.

### 2.1   Preprocessing

The proposed method includes the following preprocessing steps:

- Cropping strategy: Crop the training dataset to the non-zero region.
- Resampling method for anisotropic data: First, the images are reoriented to a unified direction. To obtain a larger receptive field during the training process, we tend to use a relatively complete patch for training. In this

way the model can capture better relative relationship between the various organs. Constrained by hardware conditions, the original image is downsampled to $160 \times 160$ for clises in the transverse section, and the spacing of inferior-superior axis is unified to 2.5. Both in-plane and out-of-plane with third-order spline interpolation.

– Intensity normalization method: First, the images is clipped to the range [-320, 320]. Then a z-score normalization is applied based on the mean and standard deviation of the intensity values[11].

## 2.2   Proposed Method

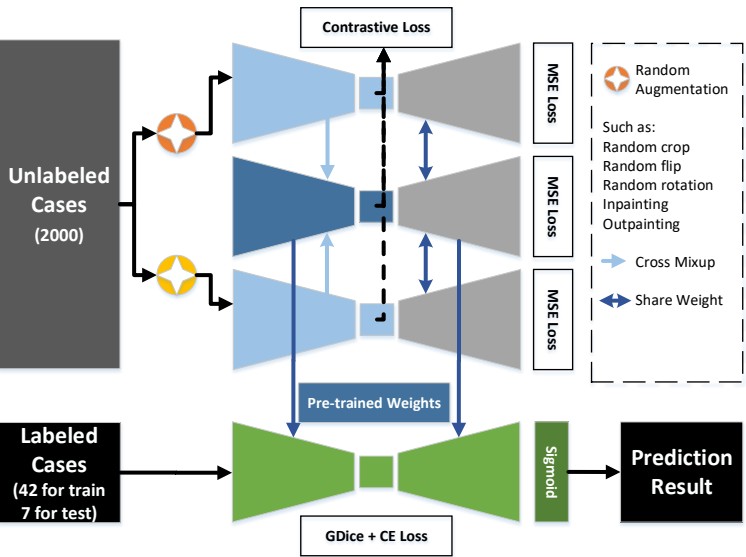

**Fig. 1.** Self-supervised pretrain and full-supervised fine-tuning framework

The unlabeled data are used to construct a self-supervised learning process to obtain a pre-trained model for augmenting the fully supervised training process. The encoder and the decoder in both pretarin stage and finetuning stage are conncected via a U shape architecture.

For the pretrain stage, the PCRL contains three different encoders and one shared decoder. The three different encoders are ordinary encoder, momentum encoder, and cross-mixup encoder, where the momentum encoder is obtained from the exponential moving average to the ordinary encode, and the cross-mixup encoder is the hybrid encoder mixed by both former encoders. Following Zhou et al.[12], for a batch of input image, different data augmentataion methods, such as random crop, random flip and random rotation are first applied to

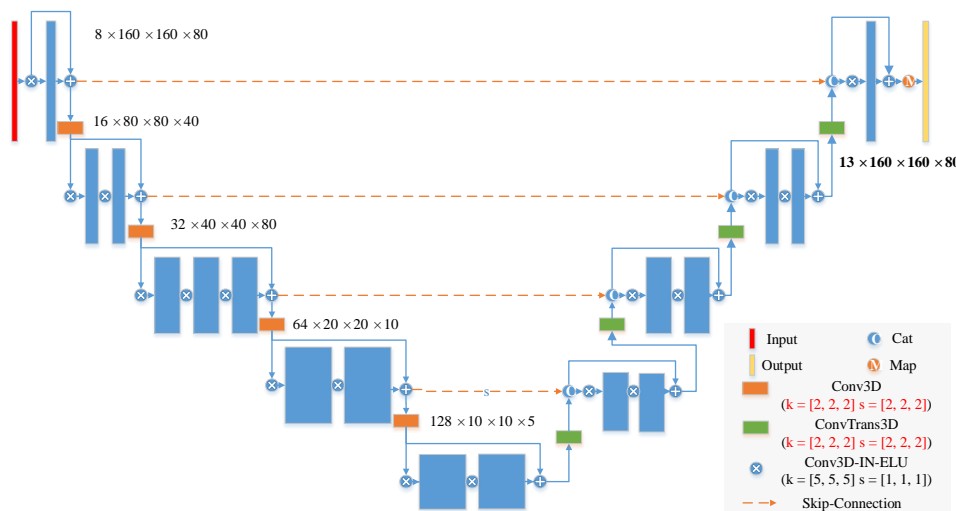

**Fig. 2.** V-Net backbone, where the input size and the number of network layers are modified accordingly to this task.

generate three batches of images corresponding to three encoders which are set as the ground truth targets of the MSE loss after decoder. Then low-level processing operations, including inpainting, outpainting are performed randomly to generate the original encoder and the momentum encoder inputs. And the input of the cross-mixup encoder is the mixup of these two inputs. The feature maps output from the original encoder and the last layer of the momentum encoder are deposited into the sequence K after global average pooling encoding to construct the constractive learning[2].

For the fintuning stage, the weights from the pre-training phase are used. And the difference is that a sigmoid layer is utilized after the decoder to perform the downstream task of segmentation.

The detail of each layer, hyper-parameters, such as stride, weight size, etc. of the backbone are shown in Fig 2

Loss function: During self-pretrain stage, the contrastive loss and MSE loss are used; During the fine-tuning stage, we use the summation between generalized Dice loss and cross entropy loss because it has been proved to be robust [5] in medical image segmentation tasks.

To reduce resource consumption, a smaller input size to reduce resource consumption. Besides, existing network frameworks (such as PyTorch) usually use full precision (Float64) for prediction. However, for intensive prediction tasks such as 3D image segmentation, the use of full-precision model parameters will greatly increase the hardware burden in the deduction process. In this work, the

half-precision (Float32) is used in the prediction stage, which reduces the GPU load by about 36% without losing the prediction accuracy.

## 3    Experiments

### 3.1    Dataset and evaluation measures

The FLARE2022 dataset is curated from more than 20 medical groups under the license permission, including MSD [9], KiTS [3,4], AbdomenCT-1K [6], and TCIA [1]. The training set includes 50 labelled CT scans with pancreas disease and 2000 unlabelled CT scans with liver, kidney, spleen, or pancreas diseases. The validation set includes 50 CT scans with liver, kidney, spleen, or pancreas diseases. The testing set includes 200 CT scans where 100 cases has liver, kidney, spleen, or pancreas diseases and the other 100 cases has uterine corpus endometrial, urothelial bladder, stomach, sarcomas, or ovarian diseases. All the CT scans only have image information and the center information is not available.

The evaluation measures consist of two accuracy measures: Dice Similarity Coefficient (DSC) and Normalized Surface Dice (NSD), and three running efficiency measures: running time, area under GPU memory-time curve, and area under CPU utilization-time curve. All measures will be used to compute the ranking. Moreover, the GPU memory consumption has a 2 GB tolerance.

### 3.2    Implementation details

**Environment settings**  The development environments and requirements are presented in Table 1.

**Table 1.** Development environments and requirements.

| | |
|---|---|
| Windows/Ubuntu version | Ubuntu 16.04.5 LTS |
| CPU | Intel(R) Xeon(R) CPU E5-2640 V3 @2.60GHz |
| RAM | 8×4GB; 2.4MT/s |
| GPU (number and type) | 4 Nvidia Geforce RTX 2080 (8G) |
| CUDA version | 11.1 |
| Programming language | Python 3.9 |
| Deep learning framework | Pytorch (Torch 1.8.1, torchvision 0.9.0) |
| Specific dependencies | V-Net[1] / PCRL[2] |

**Training protocols**  The training protocols of the baseline method is shown in Table 2. During self-supervised pretraining, random crop, random flip, random rotation, inpainting, outpainting and gaussian blur are used for constraction of contrastive learning. During the full-supervised fine-tuning, an area with a length of 80 on the axis is randomly cropped to obtain a 3D input patch of height 80 pixels, note that each patch contains at least one foreground class.

**Table 2.** Training protocols.

| | |
|---|---|
| Network initialization | "he" normal initialization |
| Batch size | 4 |
| Patch size | 80×160×160 |
| Total epochs | 2000 |
| Optimizer | Adam |
| Initial learning rate (lr) | 0.0001 |
| Learning rate decay schedule | MultiStepLR: milestones=[100, 200, 500], gamma=0.5 |
| Training time | 11.4 day (self-pretrain) + 22.5 hours (fine-tuning) |
| Loss | Contrast Loss + MSE Loss (self-pretrain); GDice Loss + CE Loss (fine-tuning) |
| Number of model parameters | 43.60M[3] |
| Number of flops | 218.7G[4] |

## 4  Results and discussion

### 4.1  Quantitative results on validation set

**Table 3.** Quantitative results on validation set in terms of DSC. The 1st row represents the method without self-pretrained, and the 2nd row represents the method with self-pretrained. (where the Liv., RK, Spl., Pan., Aor, IVC, RAG, LAG, Gal., Eso., Sto., Duo, and LK are Liver, Right Kidney, Spleen, Pancreas, Aorta, inferior vena cava, right adrenal gland, left adrenal gland, gallbladder, esophagus, stomach, duodenum, and left kidney, respectively.)

| Organ | Liv. | RK | Spl. | Pan. | Aor. | IVC | RAG | LAG | Gal. | Eso. | Sto. | Duo. | LK | Mean |
|---|---|---|---|---|---|---|---|---|---|---|---|---|---|---|
| DSC(%) | **83.10** | **67.97** | **69.26** | **49.44** | **76.14** | **64.45** | 2.00 | 4.00 | **40.03** | **39.02** | **55.45** | **40.46** | **68.65** | 50.77 |
| DSC(%) | 73.89 | 59.06 | 61.14 | 34.64 | 74.31 | 59.32 | **40.50** | **32.06** | 26.56 | 41.63 | 41.58 | 32.09 | 57.75 | 48.67 |

Table 3 illustrate the quantitative results on the provided validation set. Including the mean DSC and individual DSC for liver (Liv.), right kidney (RK), spleen(Spl.), pancreas (Pan.), aorta (Aor.), inferior vena cava (IVC), right adrenal gland (RAG), left adrenal gland (LAG), gallbladder (Gal.), esophagus (Eso.), stomach (Stm.), duodenum (Duo.) and left kidney (LK). Although all other metrics were higher than the method using the self-pretrained model. left and right adrenals were barely predictable when self-pretrained model was not used.

Table 4 illustrate the ablation study on provided 50 validation cases. Overall, the proposed method performs well on large organs such as liver and spleen, while it performs poorly on small organs such as esophageal islets and left and right adrenal glands. In addition, it can be seen that the performance of the segmentation model is significantly improved on the right and left adrenal glands after self-pretraining with unlabeled data.

**Table 4.** Ablation study on provided validation cases. The 1st row and the 3rd row represent the methods without self-pretrained, and the 2nd and 4th rows represent the methods with self-pretrained. (where the Liv., RK, Spl., Pan., Aor, IVC, RAG, LAG, Gal., Eso., Sto., Duo, and LK are Liver, Right Kidney, Spleen, Pancreas, Aorta, inferior vena cava, right adrenal gland, left adrenal gland, gallbladder, esophagus, stomach, duodenum, and left kidney, respectively.)

|  | Liv. | RK | Spl. | Pan. | Aor. | IVC | RAG | LAG | Gal. | Eso. | Sto. | Duo. | LK | Mean |
|---|---|---|---|---|---|---|---|---|---|---|---|---|---|---|
| DSC(%) | **84.28** | **61.01** | **70.49** | **50.87** | **77.94** | **67.23** | 0.00 | 0.00 | **27.41** | **46.85** | **48.95** | **39.47** | **71.79** | 49.71 |
| DSC(%) | 77.20 | 55.59 | 64.83 | 37.81 | 72.39 | 61.86 | **37.45** | **32.11** | 18.50 | 39.74 | 41.33 | 25.85 | 60.82 | 48.11 |
| NSD(%) | **70.53** | **48.56** | **59.41** | **56.08** | **77.41** | **61.38** | 0.00 | 0.00 | **21.23** | **56.16** | **46.94** | **58.73** | **66.18** | 47.89 |
| NSD(%) | 60.74 | 42.31 | 48.48 | 43.42 | 63.33 | 52.52 | **48.76** | **40.30** | 12.74 | 51.03 | 36.48 | 38.32 | 48.30 | 45.13 |

Due to memory limitations, our method uses a smaller raw input size of the network as well as a smaller channel size, which exacerbates the risk of the model losing contextual information when dealing with small targets. The process of pretraining on unlabeled data improves the upstream feature extraction part of the model for feature representation under specific data distribution, which can effectively mitigate the risk of small organ loss.

### 4.2   Qualitative results on validation set

Fig 3 present some examples on our splitted validation set. It can be found that the method using pretrained model from unlabeled data performs better for the prediction of small organs such as left and right adrenal glands compared to the method that does not utilize unlabeled data. Also, due to the use of sliding windows in our method and the preprocessing strategy of uniform spacing, there may be a certain degree of missing prediction when the input scan interval is too large or when the scan spacing differs too much from the standard spacing, which is a major reason for the decrease in evaluation metrics.

**Table 5.** Overview of DSC and NSD metrics on test set (where the Liv., RK, Spl., Pan., Aor, IVC, RAG, LAG, Gal., Eso., Sto., Duo, and LK are Liver, Right Kidney, Spleen, Pancreas, Aorta, inferior vena cava, right adrenal gland, left adrenal gland, gallbladder, esophagus, stomach, duodenum, and left kidney, respectively.)

| Organ | Liv. | RK | Spl. | Pan. | Aor. | IVC | RAG | LAG | Gal. | Eso. | Sto. | Duo. | LK | Mean |
|---|---|---|---|---|---|---|---|---|---|---|---|---|---|---|
| DSC(%) | 61.68 | 56.55 | 53.04 | 35.05 | 72.17 | 60.38 | 43.97 | 35.42 | 27.30 | 34.85 | 35.77 | 26.42 | 58.54 | 46.24 |
| NSD(%) | 41.18 | 36.67 | 33.50 | 35.37 | 60.77 | 48.89 | 56.96 | 45.05 | 16.90 | 44.19 | 26.31 | 34.08 | 41.27 | 40.09 |

### 4.3   Results on final testing set

Our final results on the test set are shown in Table 5. The final mean DSC value is 46.26%, and the mean NSD is 40.09%. The results show that the model also responds well to small targets that are difficult to segment, such as adrenal glands. The results on the test set are consistent with those of the validation set.

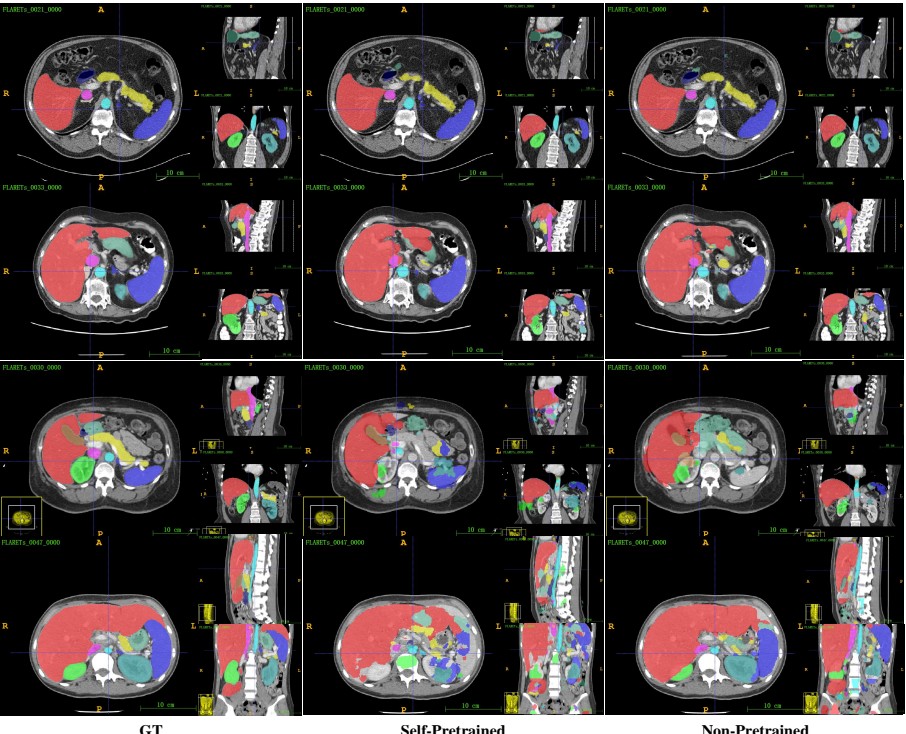

GT                              Self-Pretrained                        Non-Pretrained

**Fig. 3.** Qualitative results on some examples. First two columns are some good cases and the last two columns are some worse cases.

### 4.4   Segmentation efficiency results on validation set

To balance performance and resource consumption, We perform a scaling operation on the slicer in the transverse section, while taking a random sliding window in the Inferior-Superior axis direction to obtain a uniform size input patch. Also, the images are stretched to a fixed axis spacing of 2.5 before processing. This means that the model prediction efficiency will be greatly reduced for long range CT scans where the extent of the abdominal cavity cannot be determined (e.g., some cases in the validation set), while it is efficient for CT data where the extent of the abdominal cavity is more certain (e.g., 50 cases in the training set).

### 4.5   Limitation and future work

As mentioned before, although the sliding window strategy can effectively reduce the resource burden compared to the overall processing, it may also lead to more time-consuming and unnecessary resource wastage on CT data with larger scan ranges, and can also result in incorrect segmentation results in non-target (abdominal) intervals. In addition, the predictive power of the model for small

organs remains limited. In the future, we will focus on addressing these two aspects and exploring more possibilities for unlabeled data.

## 5    Conclusion

In this work, we proposed a method based on PCLR and V-Net to segment abdomial organs fast and cost low-resource. The self-supervised pre-trained model obtained from a large amount of unlabeled data effectively improves the prediction ability of the segmentation model for small organs such as adrenal glands. It performs well on healthy data with well-defined target intervals, however, it performs poorly and is relatively time-consuming for CT data with large scan areas.

**Acknowledgements** The authors of this paper declare that the segmentation method they implemented for participation in the FLARE 2022 challenge has not used any pre-trained models nor additional datasets other than those provided by the organizers. The proposed solution is fully automatic without any manual intervention.

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
