# OpenReview forum: "Self-pretrained V-net based on PCRL for Abdominal Organ Segmentation"
_MICCAI.org/2022/Challenge/FLARE_

### Official Review · Reviewer_V6SL · 2022-09-14
**Same as that of Paper 16: PCRL-inspired Self-supervised Training**

**Rating:** 6
**Confidence:** 4

**Review:**

# Summary

The method first performs self-supervised learning on the unlabeled data using Perservational Contrastive Representation Learning (PCRL) and finetuning the pre-trained model by performing supervised learning on the labeled data. Half precision training was used to reduce the load to the GPU with a small tradeoff to the performance.

# Strengths

- Uses self-supervised learning (PCRL) to harness the unlabeled data
- Focused to reduce the resource comparison making the maximum GPU memory used 1719 MB

# Weaknesses

- The methodology mainly covers the PCRL portion foreshadowing the actual work
- No proper analysis of results on the validation set

# Details
- Floar64, Float32, and Float16 are known as double-precision, full-precision, and half-precision. Either the author has mistakenly written Float32 for half-precision or no work was done on reducing the precision.
- In the third paragraph of **Introduction**, the citation for _residual U-net_ is wrong; it is the same as that of V-Net
- The idea of horizontal plane scaling and vertical sliding windows is not clear. Which axis do you mean by horizontal and vertical?
- The resampling method for anisotropic data is not clear about the transverse plane and axis spacing.
- The final layer for the prediction should be softmax not _sigmoid_ since it is a multi-class classification, not multi-label
- _an area with a length of 80 on the axis is randomly cropped_ no idea on what's going on here.
- The loss is Dice Loss in Table 2 but Generalized Dice in other portions of the paper
- The rows seem the same for Tables 3 and 4. A notation should be kept in the table itself rather than in the description for readers' clarity.
- The author has talked about 20 validation cases but which cases are not clear. The provided cases are 50 in number; do you mean the internal validation set?
- How can the sliding window strategy consume more time and waste unnecessary resources in comparison to whole volume input?
- No analysis for Figure 3 by the author himself and the figures are too small to be analyzed by the readers.

---

> ### Author Response · Authors · 2022-10-13
> **Thank you very much for your meticulous review, we have fully considered your suggestions and have revised the article accordingly.**
>
> 1. The 'float32' has been modified to 'float66'.
> 2. The erroneous reference here has been corrected.
> 3. In our initial description, the horizontal plane is the transverse section, and vertical is the Inferior-Superior axis. We have revised the relevant statements.
> 4. For the slicer in the transverse section, we did not perform a uniform operation on the spacing, but scaled it directly, while for the Inferior-Superior axis direction, the spacing was uniformed to 2.5.
> 5. The expression has been noted and revised.
> 6.Here, we actually use a sliding window crop in the z-axis to obtain 3D data with a height of 80, and we have modified the corresponding expression in the text.
> 7. Thanks to your careful review, this article ended up using Generalized Dice, and we have adjusted the instructions in the table.
> 8. We did not initially keep the instructions in the table given the size limitations of the table. The reviewers' concerns are well understood, and we have therefore included explanatory notes on the relevant symbols in the header of the table.
> 9. Since the labeling of the validation set was not provided, we had used the 20 examples of samples from the provided training set as the validation set not involved in the training when training the model early on, and forgot to modify this part after the article was completed. We have followed and modified this error.
> 10. The sliding window strategy requires splitting the data and splicing the results of multiple predictions, which obviously takes more time than whole volume input, which only predicts once.
> 11. The explanation of Figure 3 is explained in the original text, see Section 3.2 for details.

---

### Official Review · Reviewer_dyn4 · 2022-09-14
**the authors' results have room for further improvement.**

**Rating:** 5
**Confidence:** 4

**Review:**

Comments to the Author

In this paper, building on previous work, the authors apply self-pretrained V-net based on PCRL for abdominal organ segmentation. But I found that for the task of this paper, the authors' results have room for further improvement.

- There is extra unreadable information in the author column
- In the fourth line, "provide" should be replaced with "provided".
- In their contribution, the authors compress the input size and use a smaller network width in order to reduce resource consumption and speed up the reasoning process. Does this affect the original accuracy of the model? And how?
- What is the meaning of the lack of declaration of some of the ICONS in Fig. 1, and the "Argumentation"? Augmentation is it? Hope to give the explanation for the convenience of readers.
- Delete section 2.3.
- In Table 3, the authors use bold font for control, but it is not shown on organs such as "Gal."
- The text of this article needs further polishing.
- etc..

Please go through the paper and improve the experimental results and wording.

---

> ### Author Response · Authors · 2022-10-13
> **Thank you very much for your meticulous review, we have fully considered your suggestions and have revised the article accordingly.**
>
> 1. The author column has been revised.
> 2. Thank you for your suggestion, we have adjusted the relevant description.
> 3. As you said, reducing the size does have a negative impact on the prediction accuracy. Considering the limitation of the model size in this competition, reducing the size is the only way we can guarantee the success of the model upload, and we will try to avoid this problem in our subsequent research.
> 4. The wrong word in Figure 1 has been corrected, and correspond explanations have been added in relevant paragraph.
> 5. Section 2.3 has been deleted.
> 6. Thank you for your careful review, the issue has been revised.

---

### Official Review · Reviewer_efYj · 2022-09-16
**More weaknesses in the paper itself**

**Rating:** 4
**Confidence:** 3

**Review:**

The authors designed a preservational contrastive representation of learning (PCRL) using different argumentations on unlabeled data to get the self-pre-trained network weight.

In Sections Experiments, PCRL has only improved segmentation performance on the right adrenal gland (RAG) and left adrenal gland (LAG).

There are some problems, which must be solved before it is considered for publication. If the following problems are well-addressed:

* It is noted that your manuscript needs careful editing by someone with expertise in technical English editing paying particular attention to English grammar, spelling, and sentence structure so that the goals and results of the study are clear to the reader.

* compressing the input size and utilizing a smaller width is not available as a contribution
* The contrastive loss mentioned in your method is not clearly defined
* In Section 2.2, The detailed part of the PCRL framework is not elaborated on. For example, whether different data augmentation is used for different encoders and whether the outputs of different encoders are used directly without any regulation.
* The overall results of your method are worse and the damage to performance is not analyzed at the method level.

---

> ### Author Response · Authors · 2022-10-13
> **Thank you very much for your meticulous review, we have fully considered your suggestions and have revised the article accordingly.**
>
> As a solution to this challenge, the proposed method is only a limited attempt on the basis of PCRL. As you said, our result is not satisfactory. There may be many reasons for this result. We will find ideas and solve them by combining the winning scheme of this competition. Thank you again for your attention and review of this article.

---

### Official Review · Reviewer_GwVp · 2022-09-19
**A method based on V-Net which uses PCRL for abdominal organ segmentation**

**Rating:** 6
**Confidence:** 4

**Review:**

Inspired by retained contrastive learning, this paper proposes a V-Net, which uses unlabeled data for self pre training, and uses labeled data for fine tuning. The article has certain innovation points and high integrity, but the proposed methods did not achieve better expected results, and there is a lack of relevant discussion in some chapters, which will appear in the list of suggestions below.

Here are some suggestions for improvement:
1. Figure 3 should be adjusted, and it should not occupy a page of space.
2. The article should properly discuss the possible reasons for the decline of other organ indicators after using the self training model.
3. The article should describe the reasons for not using post-processing
4. For the loss function, the article should properly describe why to use the comparative loss and MSE loss in the self training stage.

---

> ### Author Response · Authors · 2022-10-13
> **Thank you very much for your meticulous review, we have fully considered your suggestions and have revised the article accordingly.**
>
> 1. The position of Figure 3 has been adjusted.
> 2. We have explained this in the Conclusion.
> 3. We did not try in this direction when we conceived the solution of this competition, and we will try accordingly in the subsequent research in conjunction with the winning method of this competition.
> 4. For the comparison loss and MSE loss, this scheme uses the same idea as PCRL.

---

### Comment · Reviewer_MtuJ · 2022-09-25
**In this work, the authors proposed a PCLR and V-Net-based method to segment abdominal organs for FLARE 2022 challenge with low mean DSC and NSD**

In this work, the authors proposed a PCLR and V-Net-based method to segment abdominal organs for FLARE 2022 challenge.

Strengths: A novel method based on PCLR and V-Net

Weaknesses: An Engineering method with low mean DSC and NSD.

Suggestions for authors:

1: If your preprocessing method is based on some previous work/works, please indicate that.

2: Throughout your paper, there are some English grammar mistakes and typos, such as
Selfsupervised

adopted as a practical[2], on the original V-Net[7], and PCRL[11], abdominal

3: You don’t have any postprocessing method; how you make assure that the final predictions are fine rather than coarse? Or what alternative do you have to achieve fine segmentation outcomes?

4: No clear methodology; please revisit your method and modify it accordingly.

5: Add an ablation study.

---

> ### Author Response · Authors · 2022-10-13
> **We have made a point-by-point response to the reviewer's comments and revised our manuscript**
>
> 1. Relevant citations have been added.
> 2. The relevant English grammar mistakes and typos have been corrected
> 3. We did not try in this direction when we conceived the solution of this competition, and we will try accordingly in the subsequent research in conjunction with the winning method of this competition.
> 4. Our approach is mainly based on PRCL and Vnet. We have made appropriate additions to the method section and modified some of the images.
> 5. The ablation study has been discussed and illustrated in Table 4 and Section 4.1.

---

### Meta-Review · Program_Chairs · 2022-09-28

**Recommendation:** Major Revision
**Confidence:** 5

**Metareview:**

Reviewers raise many concerns and suggestions. Please address all comments in the revised manuscript.

---

> ### Author Response · Authors · 2022-10-13
> **We have made a point-by-point response to the reviewer's comments and revised our manuscript**
>
> We have made a point-by-point response to the reviewer's comments and revised our manuscript